

# High-resolution map of sugarcane cultivation in Brazil using a phenology-based method

Yi Zheng[1], Ana Cláudia dos Santos Luciano[2,3], Jie Dong[4], Wenping Yuan[1*]

[1]School of Atmospheric Sciences, Southern Marine Science and Engineering Guangdong Laboratory (Zhuhai), Sun Yat-sen
University, Zhuhai 519082, Guangdong, China
[2]Department of Biosystems Engineering, College of Agriculture "Luiz de Queiroz" (ESALQ), University of Sao Paulo, Sao
Paulo, P.O. Box 9 Av. Padua Dias 11, 13418-900 Piracicaba-SP, Brazil
[3]Interdisciplinary Center on Energy Planning (NIPE), UNICAMP, 13083-896 Campinas, Brazil
[4]College of Geomatics & Municipal Engineering, Zhejiang University of Water Resources and Electric Power, Hangzhou
310018, Zhejiang, China

*Correspondence to*: Wenping Yuan (yuanwpcn@126.com).

**Abstract.** Sugarcane is the most important source of sugar, and its cultivation area has undergone rapid expansion, replacing

other crops, pastures, and forests. Brazil is the world's largest sugarcane producer and contributed to approximately 38.6% of

the world's total production in 2019. Sugarcane in Brazil can be harvested from April to December in south-central area and

from September to April in northeast area. The flexible phenology and harvest conditions of sugarcane in Brazil make it

difficult to identify the harvest area at state to country scales. In this study, we developed a phenology-based method to identify

the harvest area of sugarcane in Brazil by incorporating the multiple phenology conditions into a time-weighted dynamic time

warping method (TWDTW). Then, we produced annual 30-m spatial resolution sugarcane harvest maps (2016-2019) for 14

states in Brazil (over 98% of the harvest area) based on the proposed method by using Landsat-7/8 and Sentinel-2 optical data.

The proposed method performed well in identifying sugarcane harvest area with limited training sample data. Validations for

the 2018 harvest year displayed high accuracy, with user's, producer's, and overall accuracies of 97.00%, 88.08%, and 92.99%

in Brazil, respectively. In addition, the identified harvest area of sugarcane exhibited good correlations with the agricultural

statistical data provided by the Brazilian Institute of Geography and Statistics (IBGE) at the municipality, microregion, and

mesoregion levels. The 30-m Brazil sugarcane harvest maps can be obtained at https://doi.org/10.6084/m9.figshare.14213909

(Zheng et al., 2021).



## 1 Introduction

Sugarcane (*Saccharum officinarum*) is a semi-perennial crop that can be cut multiple times throughout several years in tropical and subtropical areas (Abdel-Rahman and Ahmed, 2008; Sindhu et al., 2016; Mulyono et al., 2017). Sugarcane is an important sugar and energy crop. Over 70% of world sugar comes from sugarcane (Brar et al., 2015; Iqbal et al., 2015). Sugarcane
ethanol is an alternative energy source that can reduce $CO_2$ emission resulting from fossil fuel use (Bordonal et al., 2015; Jaiswal et al., 2017). In addition, sugarcane leaves can be used as fodder and be manufactured into paper and bioenergy (Leal et al., 2013). Sugarcane accounts for approximately 20% of global crop production over the period from 2000–2018, and this value is almost twice the share of maize, the second most produced crop worldwide (FAO, 2020). Sugarcane has now become a crop with high socio-economic importance for Brazil and other producers, such as India, China, Thailand, Pakistan, the
United States, and Australia (Monteiro et al., 2018). Recently, sugarcane has undergone rapid expansion by occupying the land of other crops, pastures, and forests (Adami et al., 2012; Ferreira et al., 2015; Wachholz de Souza et al., 2017; Defante et al., 2018). The expansion of sugarcane production areas can influence regional land cover use, water use, greenhouse gas emission, soil carbon balance, and climate change (Loarie et al., 2011; Mello et al., 2014; Zhang et al., 2015; Jaiswal et al., 2017). Timely and accurate estimates of the distribution, harvest area and growing conditions of sugarcane are crucial for
sustainable sugarcane production and national food security.

Recently, single-date or time series moderate to high resolution remote sensing optical data (e.g., Landsat, Sentinel-2, SPOT, and CBERS) and synthetic aperture radar (SAR) data (e.g., PALSAR and Sentinel-1) have been used for crop mapping based on unsupervised and supervised classification methods (Lin et al., 2009; Johnson et al., 2014; Li et al., 2015; Belgiu et al., 2018). The most current and popular classification method is machine learning, such as random forest (Zhou et al., 2015;
Luciano et al., 2018, 2019; Wang et al., 2019), support vector machine (Johnson et al., 2014; Zheng et al., 2015), and neural networks (Cai et al., 2018). For sugarcane identification, Junior et al. (2017) mapped the area of sugarcane in Paraná state in Brazil based on supervised maximum likelihood classification using Landsat/TM/OLI and IRS/LISS-3 images. Depending on the sugarcane map from 2009-2014 generated by the Canasat Project (Rudorff et al., 2010) as ground truth sample data, a random forest classification model was calibrated at 10 sites located across São Paulo state and employed to the entire state in
2015, showing the ability to create spatial and time generalization models (Luciano et al., 2019). Jiang et al. (2019) identified sugarcane in Zhanjiang city in China using machine learning methods and Sentinel-1A/2 time series satellite data. However, these methods strongly depend on extensive training samples, which are time-consuming and labor-intensive to obtain at the state and country scales (Dong et al., 2020a; Wang et al., 2020). For example, the U.S. Department of Agriculture (USDA) National Agricultural Statistics Service (NASS) produced the 30-m resolution Cropland Data Layer (CDL) product using a
decision tree classification method, and this approach mainly relied on the large volume of USDA Common Land Unit (CLU) data to establish training samples that are updated annually (Boryan et al., 2011). Only in Nebraska did the CLD product need more than 250,000 CLU records for training the classification method.





Phenology-based algorithms are also commonly used in regional classification (Wardlow et al., 2007; Zhong et al., 2014; Massey et al., 2017; Dong et al., 2020b). These methods have been proposed and developed based on the crop calendar, e.g.,

the germination, tillering, grand growth, and ripening phases. Li et al. (2015) identified sugarcane on the Leizhou Peninsula in China by comparing the polarization features (such as scattering angle and polarization entropy) of sugarcane with those of other land use types in the early, middle, and late tillering periods using TerraSAR-X images. The result indicated that the tillering period is a suitable growing phase that can be used for sugarcane cultivation area mapping. Mulyono et al. (2017) identified sugarcane plantations in the Magetan district of East Java Province in Indonesia using support vector machine and

the crop phenology profile of enhanced vegetation index (EVI) time series derived from Landsat 8 images. Phenology-based methods would be a potential alternative approach for identifying crop cultivation areas at the country scale with a low volume of training samples (Dong et al., 2020a).

Dynamic time warping (DTW) is an effective phenology-based method in crop classification at large scales that compares the differences in seasonal variations in the vegetation index of a target crop with those of other crop types and natural vegetation.

The original DTW method was developed for speech recognition and then employed for phenology-based classification using time series satellite images (Sakoe and Chiba, 1978; Petitjean et al., 2012; Petitjean and Weber, 2014). However, the DTW neglects the temporal ranges when searching the best alignment between two time series. Maus et al. (2016) proposed a time-weighted version of the DTW method (TWDTW) by adding a temporal weight that accounts for the seasonality of crops into the DTW method to balance shape matching and phenological changes. The TWDTW method performed well in identifying

winter wheat, crop, and forest with limited training data (Maus et al., 2016; Belgiu et al., 2018; Manabe et al., 2018; Dong et al., 2020a).

Brazil is the world's largest sugarcane production country and contributed to 37.6% of the world's total harvest area in 2019, followed by India (18.9%), Thailand (6.9%), China (5.3%), Pakistan (3.9%), and Mexico (3.9%) (FAO, 2020). Sugarcane in Brazil is mainly located in south-central and northeast areas. Sugarcane is a semi-perennial crop. Its life cycle begins with the

planting of a stem cutting and grows for approximately 12 months or 12-18 months depending on the season, variety, and region of planting (Rudorff et al., 2010). Generally, in the south-central area, sugarcane can be harvested from April to December with a harvesting season spanning 9 months. In the northeast areas, sugarcane can be harvested from September to April in the next year with a harvesting season spanning 8 months. The flexible phenology and harvest conditions of sugarcane in Brazil make it difficult to identify the harvest area at large scales, particularly at the state to country scales. In this study,

based on Landsat and Sentinel-2 satellite data, we proposed a phenology-based method by incorporating multiple phenological conditions of sugarcane into the TWDTW method to automatically identify the harvest area of sugarcane in Brazil at a spatial resolution of 30-m from 2016 to 2019.



## 2 Data and methods

### 2.1 Study area

In Brazil, sugarcane is mainly located in the south-central and northeast regions, particularly in the state of São Paulo. In our study, we mapped the harvest area of sugarcane in 14 states in Brazil, including São Paulo (SP, accounting for 55.1% of the harvest area in Brazil), Goiás (GO, 9.5%), Minas Gerais (MG, 9.2%), Mato Grosso do Sul (MS, 6.8%), Paraná (PR, 6.1%), Mato Grosso (MT, 2.8%), Bahia (BA, 0.7%), Rio de Janeiro (RJ, 0.5%), and Espírito Santo (ES, 0.4%) in south-central Brazil; and Alagoas (AL, 2.8%), Pernambuco (PE, 2.4%), Paraíba (PB, 1.0%), Rio Grande do Norte (RN, 0.6%), and Sergipe (SE,

0.4%) in northeast Brazil (Fig. 1). The mapped areas constitute over 98% of the Brazilian sugarcane harvest area.

<<Figure 1>>

### 2.2 Datasets

#### 2.2.1 Satellite data

In this study, all the available Landsat-7/8 and Sentinel-2 reflectance data from July 2015 to February 2020 were used to

generate 16-day composite normalized difference vegetation index (NDVI) series at the Google Earth Engine (GEE) platform. Landsat-7 and Landsat-8 data had a 30-m spatial resolution and a 16-day temporal resolution. The quality band BQA was used to remove pixels contaminated by clouds. The Sentinel-2 data had a 10-m spatial resolution and a 5-day temporal resolution. The quality band QA60 was used to remove pixels contaminated by clouds. NDVI was computed using the reflectance data from the near infrared band (ρNIR) and red band (ρRED):

$$NDVI = \frac{\rho_{NIR} - \rho_{RED}}{\rho_{NIR} + \rho_{RED}} \tag{1}$$

The cloud-free frequencies of the 16-day composite NDVI in a year for each pixel are shown in Figure 2. For 2018 and 2019, most of the pixels had more than 20 times of good observations. For 2016 and 2017, the pixels with more than 20 times of good observations were less, while most of the pixels had more than 18 times of good observations (Fig. 2). To reconstruct the near real-time temporal profile of the 16-day NDVI, we filled the gaps in NDVI time series using a linear interpolation method

and then filtered the NDVI curves.

<<Figure 2>>

#### 2.2.2 Field data

The field sample data in this study were mainly obtained based on the high-resolution images from Google Earth. We first selected the samples using visual interpretations according to color and textures of the images from Google Earth, and we then

cross checked these samples using the NDVI series curves. Finally, we collected totally 2909 samples with 1393 for sugarcane and 1516 for non-sugarcane in the year 2018 (Fig. 1).



### 2.2.3 Agricultural statistical data

The agricultural statistical data for the sugarcane harvest area were derived from the Municipal Agricultural Production (PAM) provided by the Brazilian Institute of Geography and Statistics (Instituto Brasileiro de Geografia e Estatística – IBGE; www.ibge.gov.br). The PAM was carried out yearly across the entire country with the statistical data at the Brazil, major region, federation unit, mesoregion, microregion and municipality levels. It provides information related to the plant area and harvest area, production and average yield, and average price in the reference year for 64 agricultural products. In our study, we used the harvest area from 14 major sugarcane planted states at the federation unit, municipality, microregion, and mesoregion levels from 2016 to 2018 (the boundary lines of these regions are shown in Figure 3).

<<Figure 3>>

### 2.3 Methods

In this study, we employed a phenology-based method, namely the TWDTW method, to identify the harvest area of sugarcane in Brazil. The workflow was as follows: (1) pre-processing of the satellite dataset (i.e., Landsat 7/8 and Sentinel-2) to obtain the 16-day composite NDVI series, including low quality data removing (e.g., the elimination of clouds, cloud shadows, and Landsat 7 ETM+ Scan Line Corrector (SLC-off) gaps), NDVI compositing, NDVI gap filling and data filtering; (2) extraction of the standard NDVI curves of sugarcane in Brazil; (3) model development and sugarcane harvest area identification in Brazil based on the TWDTW method; and (4) assessment of the mapping accuracy of the Brazil sugarcane harvest area.

### 2.3.1 Time-weighted dynamic time warping (TWDTW) method

TWDTW is a time-weighted version of the DTW method for land use and land cover classification (Belgiu et al., 2018; Dong et al., 2020a). The TWDTW method performs phenology-based image classification by comparing the similarity between two temporal sequences and finds the optimal alignment, namely, the minimum distance between the two time series (Petitjean et al., 2012; Petitjean and Weber, 2014; Maus et al., 2016). The TWDTW algorithm includes three steps in land use classification and identification: (1) generate the standard curve of a selected index (e.g., NDVI) for several crops or a single crop (e.g., sugarcane) based on time series images and field sample training data; (2) find the best alignment, and generate the dissimilarities (TWDTW distances) between unknown curves of NDVI time series (i.e., the NDVI curves of the unknown pixels) with the standard NDVI curve of sugarcane; (3) identify the unknown pixels based on the TWDTW distance; in this process pixels with low distance indicate high similarity and a high probability of being associated with the specified class (i.e., sugarcane).

### 2.3.2 Employing the TWDTW method for sugarcane mapping

The growing period of sugarcane can be separated into four phases: the germination, tillering, grand growth, and ripening, and the NDVI values vary in different growing phases (Fig. 4). (1) Germination phase. Sugarcane begins to germinate





approximately 15-30 days after planting. The NDVI starts to increase in this period. (2) Tillering phase. Tillering phase starts after approximately 2 months of germination, and tillers emerge from the base of the mother shoot to form 5-10 stalks. The NDVI increases quickly in this period. (3) Grand growth phase. This phase spans a period of approximately 4-10 months after

planting. The NDVI peaks during this period. (4) Ripening and harvesting phase. In this phase, the NDVI starts to decrease, and the moisture content in sugarcane drastically drops.

From planting until the first cut, the crop is called planted sugarcane, and the growth cycle lasts between 12 and 18 months depending on the season, variety, and region of planting. After the first harvest, ratoon sugarcane is harvested yearly with a normal cycle of 12 months for a period of approximately 5 to 7 years or more (Rudorff et al., 2010). Although planted sugarcane

and ratoon sugarcane have different length in growing cycles, they shared similar NDVI curves from the grand growth to ripening phases (red symbols in Fig. 4) which can be used as the standard seasonal curve for harvest area mapping. In Fig. 4, the standard NDVI curves for sugarcane were generated by random selecting 50 sugarcane samples across Brazil from the field data in 2018 (Section 2.2.2) and calculating their averaged NDVI values in the same growing period.

<<Figure 4>>

Sugarcane in Brazil covers an extensive harvesting period (Rudorff et al. 2010). In the south-central area (including São Paulo, Goiás, Minas Gerais, Mato Grosso do Sul, Paraná, Mato Grosso, Bahia, Rio de Janeiro, and Espírito Santo), sugarcane is often harvested from April to December, with a harvesting season spanning 9 months. In the northeast area (including Alagoas, Pernambuco, Paraíba, Rio Grande do Norte, and Sergipe), sugarcane is harvested from September to April in the next year, with a harvesting season spanning 8 months. According to the phenology of sugarcane in the south-central and northeast areas,

Figure 5 shows the possible standard NDVI curves (these NDVI curves were repeated from the NDVI standard curve for sugarcane, as denoted by the red symbols and line in Fig. 4) for sugarcane in south-central and northeast Brazil. In this study, we incorporate the flexible phenological and harvest conditions of sugarcane in Brazil into the TWDTW method as follows.

<<Figure 5>>

(1) We calculated all the possible "distance" values for each pixel by comparing the seasonal changes of NDVI curve with the

standard curves of sugarcane (Fig. 5) based on the TWDTW method from 2016-2019. In this process, we can obtain multiple "distance" values for each pixel, and we selected the minimum value of the "distance" as the distance value for each pixel. (2) The "difference" between the maximum NDVI value in the growing season ($NDVI_{max}$: mean value of the two maximum NDVI in the growing season) and the minimum NDVI value in the non-growing season ($NDVI_{min}$: mean value of the two minimum NDVI in the non-growing season) was calculated for each pixel (Fig. 5). In the Brazilian Cerrado, some vegetation types or

areas (such as grassland, seasonal forest, and grazing areas) exhibited low NDVI values between August and October (from the end of the drought season to the beginning of the rainy season), and the corresponding NDVI curve is similar to the standard NDVI curve of sugarcane harvesting in that period. However, the differences in NDVI between the growing season and non-growing season for these vegetations were lower than those for sugarcane (Ferreira et al. 2004; Mueller et al. 2015). Therefore, we used the NDVI difference between the growing season and non-growing season as another criterion to distinguish sugarcane

and other vegetation types. Theoretically, pixels with large NDVI differences have a high probability of being associated with





sugarcane cultivation. (3) To achieve the highest possible identification accuracy, we used an optimization method to determine the "distance" and "difference" thresholds for identifying sugarcane. First, for each state, we set the flexible thresholds of NDVI differences ($T_{ND}$) between the growing season and non-growing season from 0.1 to the possible maximum value with an interval of 0.01. The pixels of NDVI differences larger than $T_{ND}$ were selected as the possible sugarcane pixels for further

identification. Second, we used the agricultural statistical harvest area for sugarcane at the state level to determine the "distance" threshold. A pixel with "distance" value lower than the "distance" threshold was considered a "sugarcane" pixel, and the total area of all sugarcane pixels should be equal to the statistical harvest area of sugarcane in the investigated state. (4) We optimize the $T_{ND}$ value according to the identification error by comparing the estimated area with the agricultural statistical data at the municipality level. In our study, municipalities with small areas of planted sugarcane (less than 1000 ha or less than 1% of the

total sugarcane area in the investigated state) were identified separately with other municipalities to improve the identification accuracy of the entire investigated state.

### 2.3.3 Accuracy assessment

In this study, we first assessed the identification accuracy using the selected sugarcane and non-sugarcane samples based on the high-resolution images from Google Earth (Section 2.2.2). The producer's accuracy (PA), user's accuracy (UA), and

overall accuracy (OA) were used for validation. The producer's accuracy (PA) is the percentage of surveyed reference samples correctly identified as the target class; the user's accuracy (UA) is the percentage of surveyed reference samples identified as the target class on the classification map actually confirmed by field surveys; and overall accuracy (OA) is the ratio of correctly classified samples to all the samples. Additionally, we calculated the sugarcane harvest area on the map in different administrative regions and compared them with agricultural statistical data at the municipality, microregion, and mesoregion

levels. The coefficient of determination ($R^2$) and RMAE (relative mean absolute error) between the statistical harvest area and the estimated harvest area were adopted to assess the map accuracy. The MAE (mean absolute error) can be expressed as:

$$MAE = \frac{1}{n}\sum_{i=1}^{n}\left|S_i - \hat{S}_i\right| \qquad (2)$$

where $S_i$ and $\hat{S}_i$ are the statistical area and identified area for the $i$ th administrative region, respectively. $n$ is the number of the administrative regions with valid statistical data. RMAE is the value of MAE relative to the mean value of the statistical

area for all the $n$ administrative regions:

$$RMAE = \frac{MAE}{\sum_{i=1}^{n} S_i/n} \qquad (3)$$

### 3 Results

Annual cultivation maps of sugarcane in the 14 states of Brazil from 2016 to 2019 were produced using the TWDTW method (taking 2018 as an example in Fig. 6). The validation demonstrated good performance of the proposed method in identifying

sugarcane harvest area in 2018. Based on the 2909 samples derived from Google Earth in 2018, the user's, producer's, and





overall accuracies were 97.00%, 88.08%, and 92.99% in Brazil, respectively. The performance of the method varied by state and region. For all the 14 studied states, the overall accuracy (OA) varied from 84.85% to 95.69%, with the user's accuracy (UA) ranging from 87.5% to 100% and producer's accuracy (PA) ranging from 80.70% to 93.10% for sugarcane; and the user's accuracy (UA) ranging from 80.70% to 93.10% and producer's accuracy (PA) ranging from 87.5% to 100% for non-

sugarcane (Table 1). São Paulo, the state with the largest planted area of sugarcane (accounting for over 50% of the sugarcane in Brazil), displayed high user's, producer's, and overall accuracies of 97.13%, 91.55%, and 95.4%, respectively. Goiás, the state with the second-largest planted area of sugarcane (accounting for approximately 10% of the sugarcane in Brazil), had high user's, producer's, and overall accuracies of 96.76%, 91.79%, and 95.05%, respectively. All the investigated states exhibited overall accuracy (OA) over 84% (Table 1).

<<Figure 6>>

<<Table 1>>

Additionally, the proposed method can accurately estimate the sugarcane harvest area compared with the agricultural statistical data at different administrative levels. The estimated harvest area of sugarcane in 2018 exhibited good correlations with the agricultural statistical area data derived from PAM at the municipality, microregion, and mesoregion levels. The coefficients

of determination ($R^2$) were 0.84 (N=3637), 0.95 (N=369), and 0.97 (N=87) at the municipality, microregion, and mesoregion levels, respectively, and the respective RMAEs were 38.9% (MAE=$0.11\times10^4$ ha), 23.8% (MAE=$0.64\times10^4$ ha), and 18.1% (MAE=$2.05\times10^4$ ha) (Fig. 7). The performance was better when the validated regions were aggregated to larger areas, namely the accuracy from low to high were at the municipality, microregion, and mesoregion levels.

<<Figure 7>>

The correlations between the agricultural statistical and the estimated harvest areas varied by state and region. At the municipality level, the coefficient of determination ($R^2$) between the agricultural statistical and the estimated harvest areas ranged from 0.61 to 1 in south-central Brazil and from 0.58 to 0.87 in northeast Brazil (Figs. 8-9); the RMAE ranged from 9.0% to 71.4% in south-central Brazil and from 43.4% to 58.9% in northeast Brazil (Fig. 8; Fig. 10). At the microregion level, the $R^2$ between the agricultural statistical and the estimated harvest areas ranged from 0.75 to 1 in south-central Brazil and

from 0.67 to 0.99 in northeast Brazil (Fig. 9); the RMAE ranged from 6.9% to 53.8% in south-central Brazil and from 19.5% to 48.2% in northeast Brazil (Fig. 10). At the microregion level, the $R^2$ between the agricultural statistical and the estimated harvest areas ranged from 0.73 to 1 in south-central Brazil and from 0.99 to 1 in northeast Brazil (Fig. 9); the RMAE ranged from 5.4% to 51.3% in south-central Brazil and from 6.1% to 29.0% in northeast Brazil (Fig. 10). Validation at all three levels showed high performance, with high $R^2$ and slope close to 1. In general, Mato Grosso and Bahia in the northern part of south-

central Brazil and Sergipe in the southern part of northeastern Brazil displayed lower $R^2$ and higher RMAE values (Figs. 8-10). The performance for São Paulo and Goiás were higher than those for most other states, except Rio de Janeiro (Figs. 8-10). Finally, we assessed the capability of the method and standard seasonal changes in NDVI (i.e., standard NDVI curves) acquired from a single year (2018) to apply them to other years (2016 and 2017). Results indicated the $R^2$ and RMAE values for the period of 2016–2018 changed little in most states (Figs. 9-10).



<<Figures 8-10>>

**4 Discussion**

As the largest global producer of sugarcane, Brazil contributed to approximately 38.6% of the world's sugarcane production in 2019 and played an important role in retaining the global demand for sugarcane (FAO, 2020). While the cultivation area of sugarcane in Brazil has increased by approximately 11% over the past 10 years according to census data (FAO, 2020), which

indicated substantial land cover changes and important feedback to regional climate systems (Loarie et al., 2011; Mello et al., 2014; Jaiswal et al., 2017). The first harvest area map of sugarcane in Brazil at a 30-m spatial resolution was generated only for São Paulo state based on an automatic image classification method (Rudorff et al., 2005). Subsequently, the cultivation map of sugarcane was updated using visual/manual image interpretation through 2013, and the coverage of the map extended from São Paulo state to a total of 8 states in the south-central region of Brazil, accounting for 88% of the sugarcane cultivation

area in Brazil, as reported by the Canasat Project (Rudorff et al., 2010). Souza et al. (2020) reconstructed annual land use and land cover information at a 30-m spatial resolution from 1985-2017 for Brazil based on random forest method and Landsat data trained by plenty of samples selected from existing land cover maps, which provides the cultivation map of sugarcane belonging to a subclass of agriculture (the producer's and user's accuracies for agriculture were 83.3% and of 81.3%, respectively). However, these prevailing methods strongly require large volume of training samples, which makes it difficult

to update annually at large scales.

In this study, we proposed a phenology-based sugarcane classification method by incorporating multiple phenological conditions of sugarcane into the TWDTW method. Then, we identified the harvest area of sugarcane with a spatial resolution of 30-m in Brazil from 2016 to 2019 using 16-day composite NDVI series derived from Landsat and Sentinel-2 data. Our proposed method can automatically identify the sugarcane areas with limited training data and needs only one standard NDVI

curve (Fig. 4) for all the 14 sugarcane planted states in Brazil. Validations against field sample data and agricultural statistical area data showed that the generated sugarcane harvest maps were in high accuracy. For example, validations for 2018 displayed the user's, producer's, and overall accuracies of 97.00%, 88.08%, and 92.99% in Brazil, respectively.

Although our method can effectively and accurately identify the sugarcane harvest area from the regional to national or continental scales, there are still several potential uncertainties in the identification process. First, because of the quality of the

processed NDVI series, the speckle "salt-and-pepper" effects/noises exist in some areas of the harvest map. According to the map statistics in 2018, sugarcane patches with only one pixel account for 0.55% (São Paulo) to 8.6% (Bahia) of the sugarcane area (Fig. 11). In the future, the object-based identification by segmenting images into homogeneous objects, instead of the pixel-based method used in our study, may alleviate the "salt-and-pepper" effects and improve the identification performance (Belgiu et al., 2018).

<<Figure 11>>



Second, the identification accuracy for Bahia state was lower than that for other states. Because of the different harvest seasons in south-central and northeast Brazil, we identified sugarcane in the south-central and northeast areas using different phenological characteristics and standard curve combinations (Fig. 5). Bahia is a transition state between south-central and northeast Brazil. Namely, sugarcane in southern Bahia has a harvest season similar to that in south-central Brazil, and

sugarcane in northern Bahia has a harvest season similar to that in northeast Brazil. In our study, we treated Bahia as a state in south-central Brazil with a harvest season from April to December (Fig. 5), which may introduce errors to the identification in the northern part of Bahia.

Third, Mato Grosso state exhibited a lower $R^2$ and higher RMAE than other states when comparing the identified sugarcane harvest area with the agricultural statistical sugarcane harvest area. Two major biomes are located in Mato Grosso, including

the humid tropical forests of the Amazon in the north and the heterogeneous Cerrado area (a tropical savanna) in the south-central part of the state (Kastens et al. 2017). In Mato Grosso, sugarcane may be misclassified with some kinds of grassland, grazing areas, or seasonal forest, which exhibit phenological changes similar to those of sugarcane (Mueller et al. 2015; Bendini et al. 2019). The NDVI values of these vegetation types decrease between August and October (from the end of the drought season to the beginning of the rainy season) and increased thereafter (Ferreira et al. 2004), which is quite similar to that in the

harvesting stage of sugarcane, resulting in misclassification with the sugarcane harvested from August to October. In our study, we used the "difference" between the maximum NDVI value in the growing season and the minimum NDVI value in the non-growing season (see method in Section 2.3.2) to alleviate the misclassification at some extent because the "difference" for the abovementioned vegetation types is generally lower than that for sugarcane (Ferreira et al. 2004). In the future, incorporating more complex spectral-temporal variability metrics, such as the combination of more spectral information instead of NDVI

and a longer time window for several harvest seasons instead of one harvest season may help improve model performance (Mueller et al. 2015).

## 5 Data availability

The 30-m Brazil sugarcane harvest area dataset from 2016-2019 is available at https://doi.org/10.6084/m9.figshare.14213909 (Zheng et al., 2021). The dataset is provided in .tif format with pixel values of 1 for sugarcane and 0 for non-sugarcane.

## 6 Conclusion

Brazil is the world's largest sugarcane producer and contributes to approximately 38.6% of total global sugarcane production. Based on the available Landsat and Sentinel images, we produced sugarcane harvest maps with a 30-m spatial resolution (2016-2019) by incorporating multiple phenological conditions of sugarcane in Brazil into the TWDTW method. The proposed method can automatically identify sugarcane harvest area with limited training sample data. Based on 2909 samples derived

from Google Earth, the validation experiment reflected high accuracy across the 14 sugarcane planted states in Brazil, with



the user's, producer's, and overall accuracies of 97.00%, 88.08%, and 92.99% in Brazil, respectively. Additionally, the identified harvest area of sugarcane exhibited a good correlation with the agricultural statistical area data derived from PAM at the municipality, microregion, and mesoregion levels. The maps can be used to monitor the harvest area and yield of sugarcane, and evaluate the feedback to regional climate.

**Author contributions.** Wenping Yuan and Yi Zheng designed the research, performed the analysis, and wrote the paper; Ana Cláudia dos Santos Luciano and Jie Dong edited and revised the manuscript.

**Competing interests.** The authors declare that they have no conflict of interest.

**Acknowledgements**

This study was supported by the National Science Fund for Distinguished Young Scholars (41925001), National Natural
Science Foundation of China (41971018 and 31930072), and Fundamental Research Funds for the Central Universities (19lgjc02).

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





**Tables**

**Table 1: Confusion matrix of the sugarcane harvest map for the 14 states in Brazil in 2018.**

| State | | Reference | Map | | Producer's accuracy | User's accuracy | Overall accuracy |
|---|---|---|---|---|---|---|---|
| | | | Sugarcane | Non-sugarcane | | | |
| South-central | São Paulo (SP) | Sugarcane | 271 | 25 | 91.55% | 97.13% | 95.40% |
| | | Non-sugarcane | 8 | 413 | 98.10% | 94.29% | |
| | Goiás (GO) | Sugarcane | 179 | 16 | 91.79% | 96.76% | 95.05% |
| | | Non-sugarcane | 6 | 243 | 97.59% | 93.82% | |
| | Minas Gerais (MG) | Sugarcane | 95 | 17 | 84.82% | 95.00% | 89.05% |
| | | Non-sugarcane | 5 | 84 | 94.38% | 83.17% | |
| | Mato Grosso do Sul (MS) | Sugarcane | 83 | 10 | 89.25% | 100.00% | 95.02% |
| | | Non-sugarcane | 0 | 108 | 100.00% | 91.53% | |
| | Paraná (PR) | Sugarcane | 84 | 14 | 85.71% | 94.38% | 93.65% |
| | | Non-sugarcane | 5 | 196 | 97.51% | 93.33% | |
| | Mato Grosso (MT) | Sugarcane | 85 | 17 | 83.33% | 97.70% | 90.59% |
| | | Non-sugarcane | 2 | 98 | 98.00% | 85.22% | |
| | Bahia (BA) | Sugarcane | 56 | 13 | 81.16% | 100.00% | 88.60% |
| | | Non-sugarcane | 0 | 45 | 100.00% | 77.59% | |
| | Rio de Janeiro (RJ) | Sugarcane | 81 | 6 | 93.10% | 100.00% | 95.68% |
| | | Non-sugarcane | 0 | 52 | 100.00% | 89.66% | |
| | Espírito Santo (ES) | Sugarcane | 63 | 12 | 84.00% | 100.00% | 89.66% |
| | | Non-sugarcane | 0 | 41 | 100.00% | 77.36% | |
| Northeast | Alagoas (AL) | Sugarcane | 54 | 8 | 87.10% | 98.18% | 91.35% |
| | | Non-sugarcane | 1 | 41 | 97.62% | 83.67% | |
| | Pernambuco (PE) | Sugarcane | 57 | 6 | 90.48% | 100.00% | 94.64% |
| | | Non-sugarcane | 0 | 49 | 100.00% | 89.09% | |
| | Paraíba (PB) | Sugarcane | 46 | 11 | 80.70% | 92.00% | 84.85% |
| | | Non-sugarcane | 4 | 38 | 90.48% | 77.55% | |
| | Rio Grande do Norte (RN) | Sugarcane | 42 | 4 | 91.30% | 87.50% | 88.76% |
| | | Non-sugarcane | 6 | 37 | 86.05% | 90.24% | |
| | Sergipe (SE) | Sugarcane | 31 | 7 | 81.58% | 96.88% | 88.89% |
| | | Non-sugarcane | 1 | 33 | 97.06% | 82.50% | |

## 450  Figures

![Figure 1 map of Brazil showing sugarcane study areas]

**Figure 1: Study areas in Brazil for sugarcane harvest area identification, including 9 states in south-central Brazil (São Paulo, Goiás, Minas Gerais, Mato Grosso do Sul, Paraná, Mato Grosso, Bahia, Rio de Janeiro, and Espírito Santo) and 5 states in northeast Brazil**
**(Alagoas, Pernambuco, Paraíba, Rio Grande do Norte, and Sergipe), which account for over 98% of the sugarcane harvest area in Brazil. The dots represent the samples used for validation. The administrative boundary data were derived from the Brazilian Institute of Geography and Statistics (Instituto Brasileiro de Geografia e Estatística – IBGE; www.ibge.gov.br).**

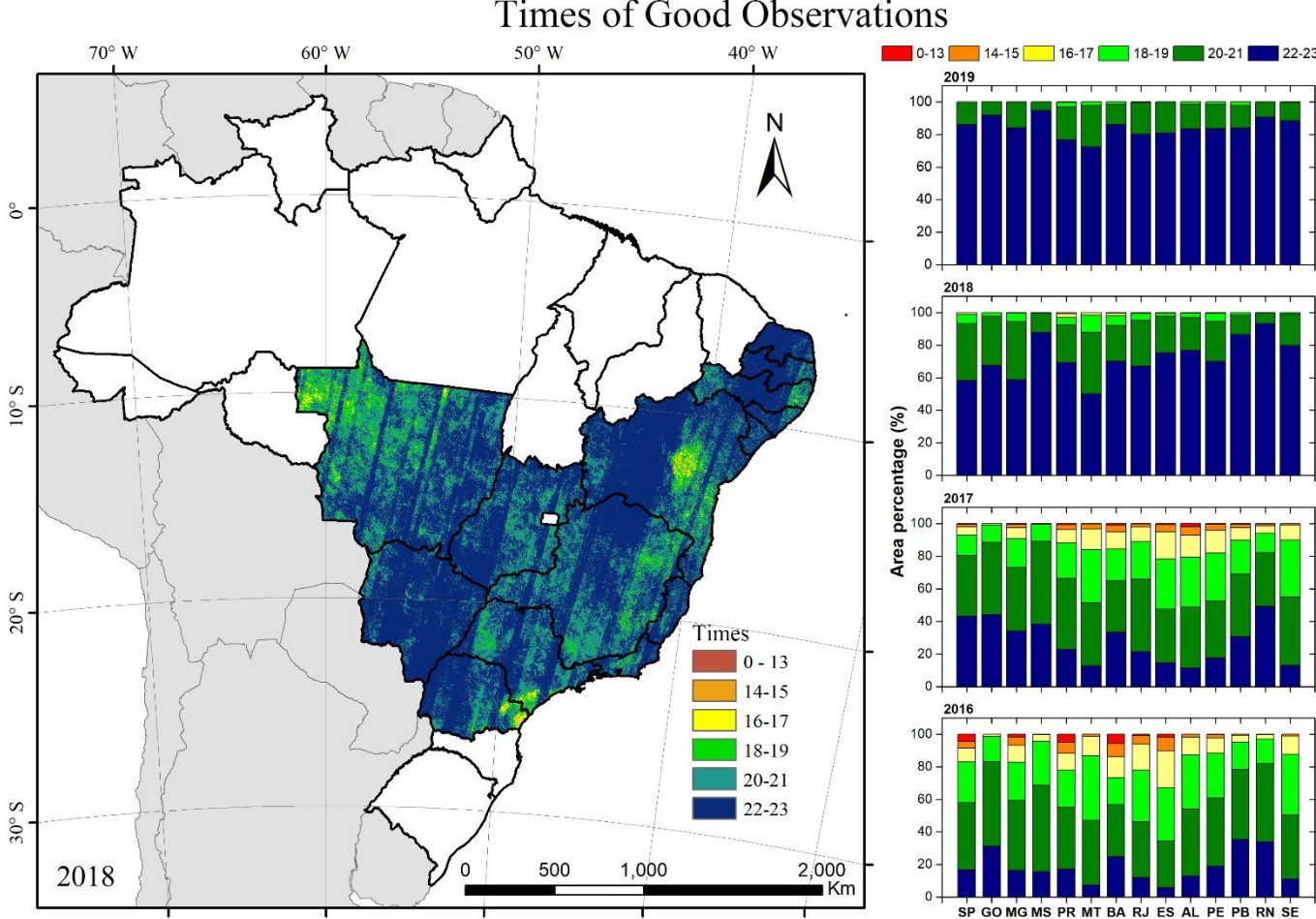

**Figure 2: Times of good observations for the 16-day composite satellite data over a year. (left) Times of good observations in 2018; (right) Area percentages of the times of good observations for each state from 2016-2019.**



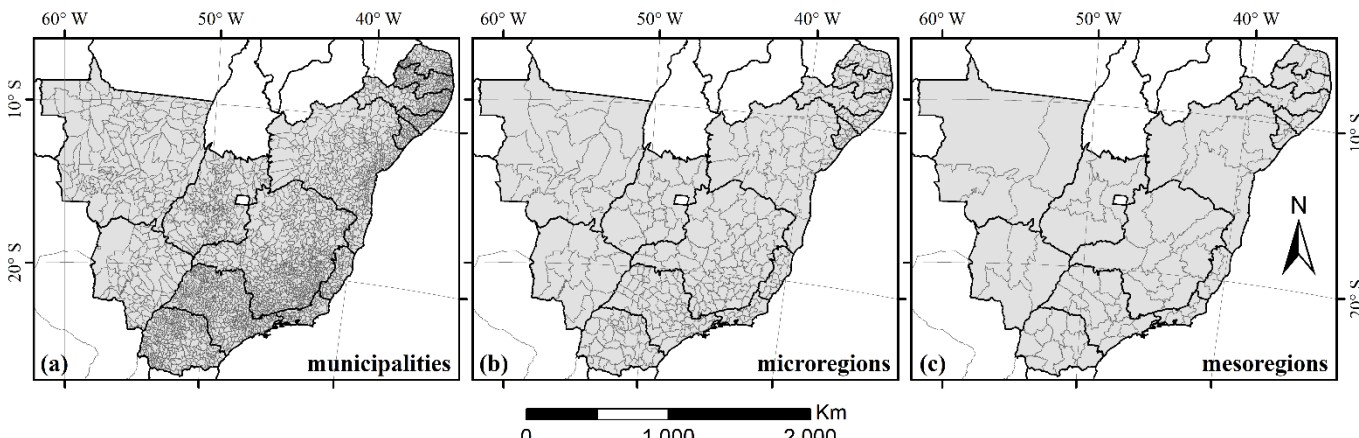

**Figure 3: The administrative boundaries for (a) municipalities, (b) microregions, and (c) mesoregions. The boundary lines at the municipality and state levels were downloaded directly from the IBGE, and we aggregated the municipalities into microregions and mesoregions according to the regions in PAM denoted by the IBGE.**

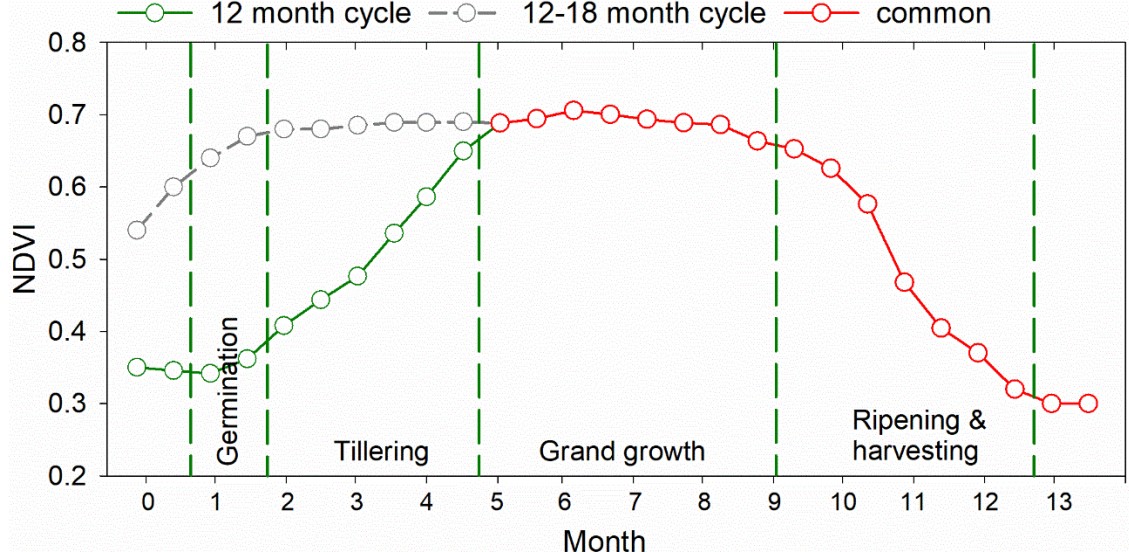

**Figure 4: Growing stages of sugarcane and the respective NDVI changes. The gray and red symbols represent the NDVI curve for the planted sugarcane with a 12–18 month cycle, and the green and red symbols represent the NDVI curve for the ratoon sugarcane with a 12 month cycle. The growing stages of the 12 month cycle sugarcane (germination, tillering, grand growth, and ripening) are labeled in the figure.**



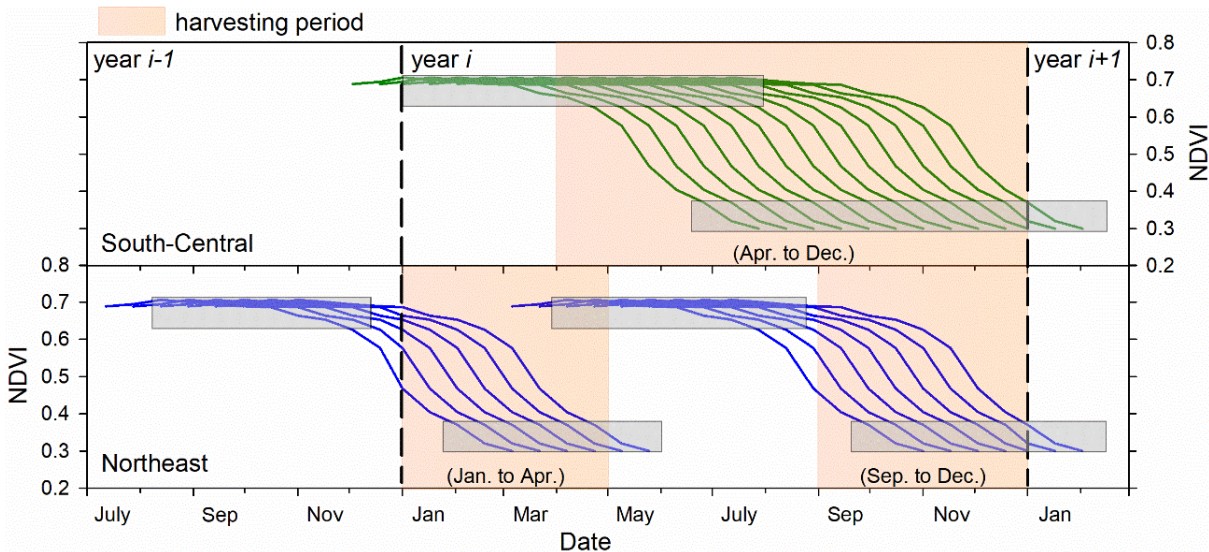

**Figure 5: Seasonal changes in the NDVI series for sugarcane in the south-central and northeast Brazil. The gray areas are the periods used to calculate the maximum NDVI value in growing season (NDVI$_{max}$) and the minimum NDVI value in non-growing season (NDVI$_{min}$).**

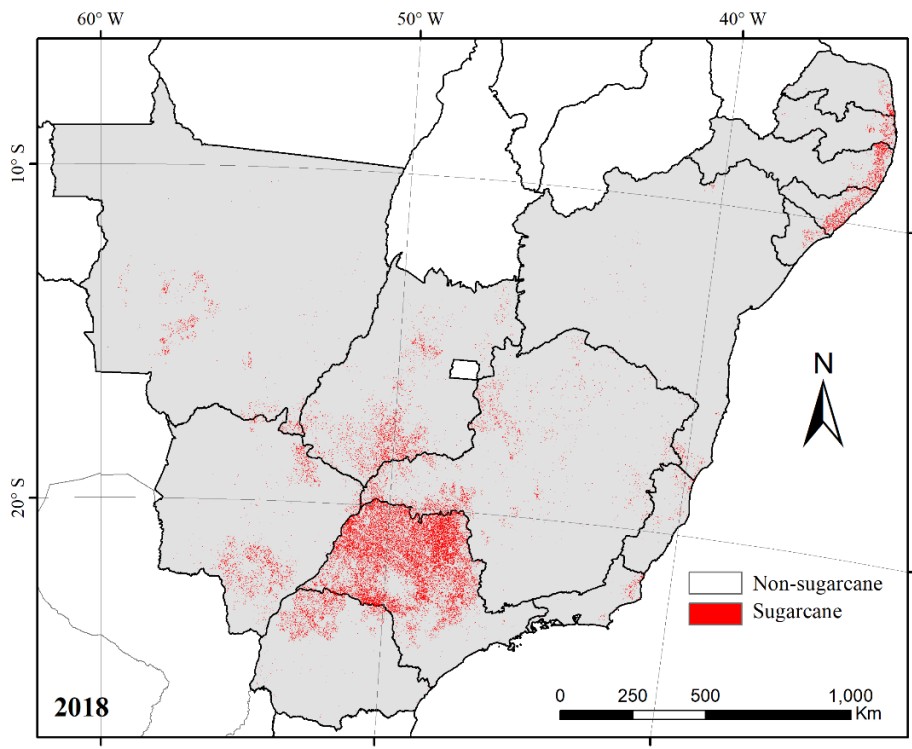

**Figure 6: Sugarcane harvest map for the 14 studied states in Brazil in 2018. The administrative boundary data were obtained from the IBGE.**


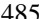


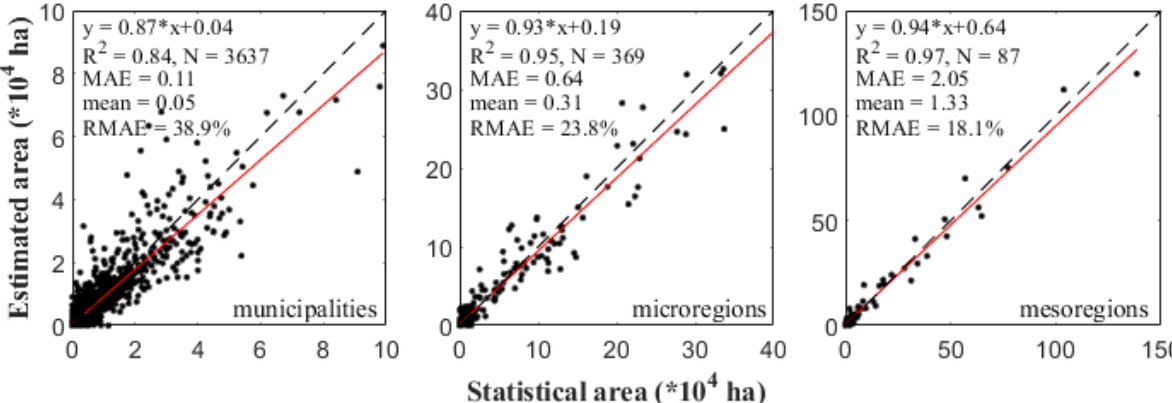

**Figure 7: Comparisons between the agricultural statistical harvest area and the estimated harvest area of sugarcane at the municipality level in Brazil in 2018. "N" and "mean" represents the number and mean value of all the valid statistical data in each figure, respectively.**




**Figure 8: Comparisons between the agricultural statistical harvest area and the estimated harvest area of sugarcane at the municipality level in the 14 states in 2018. "N" and "mean" represents the number and mean value of all the valid statistical data in each figure, respectively.**


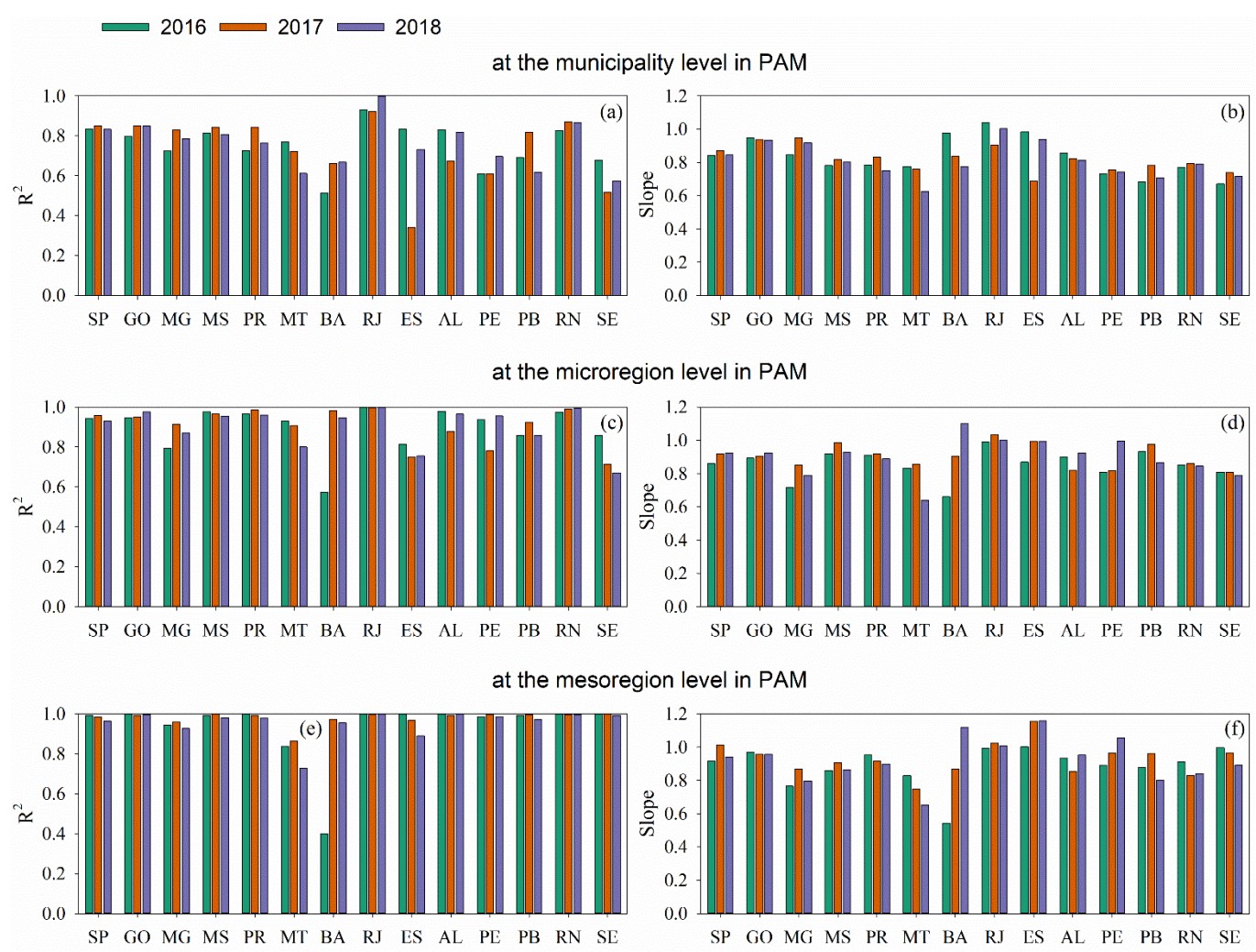

**Figure 9: Comparisons between the agricultural statistical harvest area and the estimated harvest area of sugarcane ($R^2$ and slope) at the (a-b) municipality, (c-d) microregion, and (e-f) mesoregion levels from 2016-2018.**




**Figure 10: Comparisons between the agricultural statistical harvest area and the estimated harvest area of sugarcane (MAE and RMAE) at the (a-b) municipality, (c-d) microregion, and (e-f) mesoregion levels from 2016-2018.**


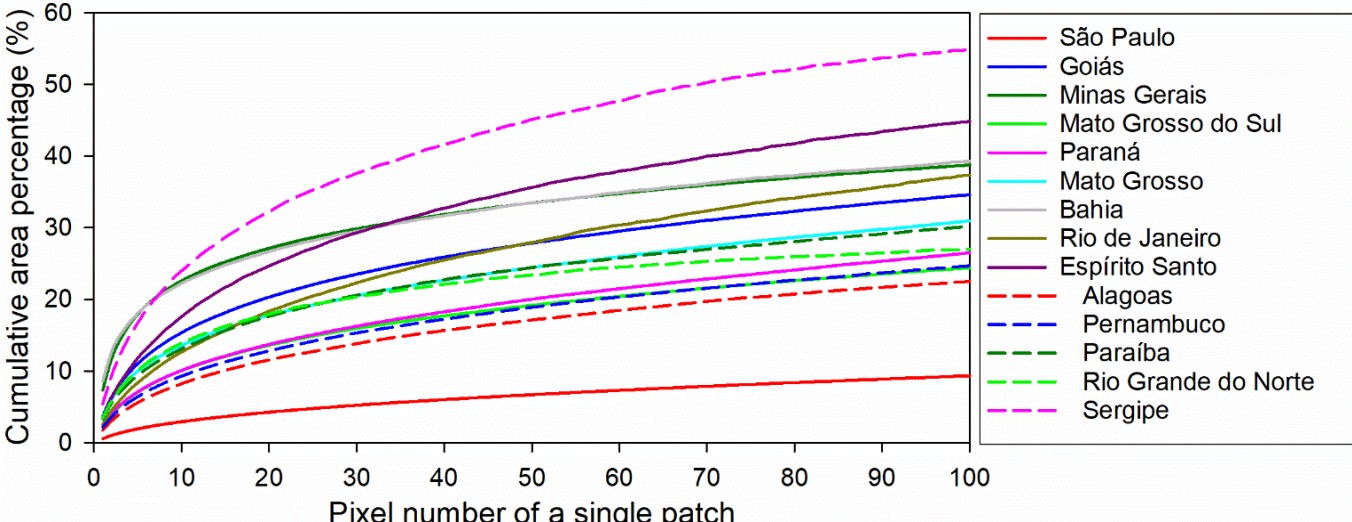

**Figure 11: Statistics for patches with different pixel numbers in the sugarcane harvest map for the 14 states in Brazil in 2018.**