# Peer review of "High-resolution map of sugarcane cultivation in Brazil using a phenology-based method"

_Earth System Science Data, 2021_

## Referee Comment (RC2)

Brazil is the world's largest sugarcane producer, and accurate estimates of the distribution, harvest area of sugarcane are crucial for sustainable sugarcane production and national food security. The study "High-resolution map of sugarcane cultivation in Brazil using a phenology-based method" aims to identify the harvest area of sugarcane in Brazil by a phenology-based method (TWDTW). The annual 30-m spatial resolution sugarcane harvest maps (2016-2019) for 14 states in Brazil have been generated with very high accuracy (over 98%).

In terms of the method, I have some concerns.

1. The field sample data.

The authors used the Google Earth images to select the sugarcane samples without any ground truth samples as references, so it is difficult to tell the crop types. Meanwhile, the authors did not describe the rules of visual interpretations. It is hard to tell the accuracy of the sugarcane samples selected.

In addition, the NDVI curves were used to select sugarcane samples, but later the selected sugarcane samples were used to extract the standard sugarcane NDVI curves to develop algorithms for sugarcane mapping. Therefore, I am confused that there was a little self-proof and not sure about the accuracy estimated based on these samples.

As these samples were selected by Google Earth images rather than filed truth data, it is not reasonable to call these samples as "filed data".

2. The TWDTW method for sugarcane mapping

In this work, the "difference" between the maximum NDVI value in the growing season (NDVImax: mean value of the two maximum NDVI in the growing season) and the minimum NDVI value in the non-growing season (NDVImin: mean value of the two minimum NDVI in the non-growing season) was calculated for each pixel. The "difference" as a criterion to map the sugarcane in Brazil. The evidence was shown as "the differences in NDVI between the growing season and non-growing season for these vegetations were lower than those for sugarcane" in lines of 177-178. Here, the vegetations compared mainly include grassland, seasonal forest, and grazing areas as shown in line of 175.

My main concern is that the difficulty of sugarcane mapping is to separate sugarcane from other crop types within agricultural system rather than vegetations in other ecosystems such as grassland and forest. This evidence cannot prove the criterion used in the sugarcane mapping is robust.

3. The "distance" and "difference" thresholds for identifying sugarcane

See lines 185-190, the agricultural statistical harvest area for sugarcane at the state and municipality levels have been used to determine the "distance" threshold and optimize the "difference" threshold, however, the statistical data were used to compare with the resultant sugarcane areas. To some extent, this approach is self-proof.

What's more, statistical data cannot present the spatial patterns of the sugarcane fields. Therefore, I doubt the ability of this approach to describe the spatial information of the sugarcane distribution in Brazil.

Minor:

1. Time-weighted dynamic time warping (TWDTW) method

TWDTW is the method used in this work to map sugarcane, but the description of the method is not detail.

2. Sugarcane in Brazil covers an extensive harvesting period, how were all the possible standard NDVI curves collected in Figure 5?

3. It is needed to provide more details about the method. In addition, more descriptions are needed about the flexible phenological and harvest conditions of sugarcane in the TWDTW in lines of 165-167.

4. In lines 170, how the possible "distance" values have been calculated? Please give more explanations here.

---

## Author Comment (AC1)

**Journal:** ESSD

**Title:** High-resolution map of sugarcane cultivation in Brazil using a phenology-based method

**MS No.:** essd-2021-88

**MS Type:** Data description paper

Dear Prof. Kirsten Elger and reviewers,

We are very grateful to you and the reviewers for the time and constructive comments on our manuscript "High-resolution map of sugarcane cultivation in Brazil using a phenology-based method" (MS No.: essd-2021-88). The comments have helped improve the paper quite tremendously.

We have carefully studied these comments by Reviewer#1 and Reviewer#2, and revised our manuscript accordingly. Please find the point-by-point responses to the comments of the two reviewers. Please note that the comments from the reviewers are in **bold** followed by our responses in regular text. The changes in our manuscript are underlined with red.

We believe the quality of the manuscript can now meet the high standard of ESSD and deeply appreciate your consideration of our manuscript.

Sincerely,

Yi Zheng, Wenping Yuan

School of Atmospheric Sciences,

Sun Yat-sen University, Zhuhai 519082, Guangdong, China

Email: yuanwpcn@126.com

**Response to Reviewer#1:**

**The paper aims to map the sugarcane in several Brazilian states based on time series and phenological conditions. The manuscript is well written, and the methodology is clearly presented. I have minor recommendations to improve the manuscript:**

Thank you for your positive comments. Your suggestions are very useful for us to improve our research. We revised our manuscript according to your comments. The changes in our manuscript are underlined with red. We believe our manuscript improved a lot after the modification. Please see the response below.

**1. L37-40. Yes, it can influence these aspects. But how? The phrase is a bit vague. The consequences of a non-sustainable sugarcane production must be briefly described to show the importance of sugarcane mapping.**

**Lines 37-40 is "The expansion of sugarcane production areas can influence regional land cover use, water use, greenhouse gas emission, soil carbon balance, and climate change (Loarie et al., 2011; Mello et al., 2014; Zhang et al., 2015; Jaiswal et al., 2017). Timely and accurate estimates of the distribution, harvest area and growing conditions of sugarcane are crucial for sustainable sugarcane production and national food security."**

Thank you. As your suggestion, we added detailed description of the influence of sugarcane expansion on these aspects in the revised manuscript:

"The expansion of sugarcane production areas can influence regional land cover use, water use, greenhouse gas emission, soil carbon balance, and climate change (Loarie et al., 2011; Mello et al., 2014; Zhang et al., 2015; Jaiswal et al., 2017). On the one hand, the replacement of other crops with sugarcane may directly affect on agriculture and food security (Mello et al., 2014; Jaiswal et al., 2017). On the other hand, sugarcane expansion may influence the local climate by altering surface albedo and evapotranspiration (Loarie et al., 2011). Additionally, sugarcane has a large amount of water requirement and is often planted in the areas where water is limiting, therefore, sugarcane expansion may cause concerns about water security (Zhang et al., 2015). Timely and accurate estimates of the distribution, harvest area and growing conditions of sugarcane are crucial for sustainable sugarcane production and national food security." (Lines 37-44 in the revised manuscript)

**2. L59-67. Only the positive aspects of those methods were reported. The authors should point out why they are still not enough to perform an ideal mapping.**

**Lines 59-67 is "These methods have been proposed and developed based on the crop calendar, e.g., the germination, tillering, grand growth, and ripening phases. Li et al. (2015) identified sugarcane on the Leizhou Peninsula in China by comparing the polarization features (such as scattering angle and polarization entropy) of sugarcane with those of other land use types in the early, middle, and late tillering periods using TerraSAR-X images. The result indicated that the tillering period is a suitable growing phase that can be used for sugarcane cultivation area mapping. Mulyono et al. (2017) identified sugarcane plantations in the Magetan district of East Java Province in Indonesia using support vector machine and the crop phenology profile of enhanced vegetation index (EVI) time series derived from Landsat 8 images. Phenology-based methods would be a potential alternative approach for identifying crop cultivation areas at the country scale with a low volume of training samples (Dong et al., 2020a)."**

Sorry for neglecting. We have added the limitation of traditional phenology-based methods in the revised manuscript:

"Phenology-based algorithms are also commonly used in regional classification (Wardlow et al., 2007; Zhong et al., 2014; Massey et al., 2017; Dong et al., 2020b). These methods have been proposed and developed based on the crop calendar, e.g., the germination, tillering, grand growth, and ripening phases. Li et al. (2015) identified sugarcane on the Leizhou Peninsula in China by comparing the polarization features (such as scattering angle and polarization entropy) of sugarcane with those of other land use types in the early, middle, and late tillering periods using TerraSAR-X images. The result indicated that the tillering period is a suitable growing phase that can be used for sugarcane cultivation area mapping. Mulyono et al. (2017) identified sugarcane plantations in the Magetan district of East Java Province in Indonesia using support vector machine and the crop phenology profile of enhanced vegetation index (EVI) time series derived from Landsat 8 images. Phenology-based methods would be a potential alternative approach for identifying crop cultivation areas at the country scale with a low volume of training samples (Dong et al., 2020a). However, these studies were developed based on several phenological thresholds (e.g., green-up date, senescence date, and length of growing season) of crops in different growing stages, which may vary by region and year and need to be calibrated when extended to other regions or years. Therefore, traditional phenology-based methods are still insufficient to perform ideal mapping and may be limited by multiple thresholds when applied to large scales, such as the country to continental scales." (Lines 62-75 in the revised manuscript)

**3. L169-190. A flowchart summarizing this process would be good.**

Good idea, a flowchart can help to describe the method more clearly. We have added a flowchart in the method section (2.3.2 Employing the TWDTW method for sugarcane mapping) as following:

"The workflow by employing the TWDTW method for sugarcane harvest area mapping are shown in Fig.5." (Line 165-166 in the revised manuscript)

[Figure]

Figure 5. The workflow by employing the TWDTW method for sugarcane harvest area mapping.

**4. L193-194. What percentage of samples used for training and validation? Did the authors perform k-fold?**

**Lines 193-194: In this study, we first assessed the identification accuracy using the selected sugarcane and non-sugarcane samples based on the high-resolution images from Google Earth (Section 2.2.2).**

We used 50 samples randomly selected from all the 1393 sugarcane samples for training to produce the standard NDVI curve for sugarcane (Lines 156-158 in the original manuscript). In the revised manuscript, we used the remaining 1343 sugarcane samples and 1516 non-sugarcane samples for

validation. We did not perform the k-fold for training and validation because we used the sugarcane samples to extract the averaged temporal pattern (standard NDVI curves) of sugarcane to develop the method. The standard NDVI curves of sugarcane exhibit similar when using different samples. In addition, performing k-fold training and validation is time-consuming when calculate TWDTW distance for each pixel.

Lines 156-158 in the original manuscript "In Fig. 4, the standard NDVI curves for sugarcane were generated by randomly selecting 50 sugarcane samples across Brazil from the field data in 2018 (Section 2.2.2) and calculating their averaged NDVI values in the same growing period."

**5. Methodology – Any post-processing was used to avoid salt-and-pepper effect?**

We did not use post-processing to avoid salt-and-pepper effect. On the one hand, as we know, there is no good method to eliminate salt-and-pepper effect. Smoothing at spatial extent may lose some detailed information on the sugarcane maps, such as roads, and ridges between the fields (see the response to comment #6). One the other hand, it is difficult to discriminate noises from true signals to smooth the sugarcane maps because some areas is highly heterogeneous and fragmented with parcels about or smaller than 30m.

**6. Results – I recommend the inclusion of a new figure, where some example areas are shown from a finer scale (zoom in). In this case, readers will be able to see the visual mapping quality in terms of delineation, presence/absence of salt-and-pepper effect, etc.**

Good idea. As your suggestion, we inserted a new figure to show the maps in a finer scale. In Fig. 9, we selected four typical areas (area A-D in Fig. 8) to zoom in and compared the sugarcane maps with images from Google Earth. We added the corresponding information in the revised manuscript.

The results section:

[revised manuscript text omitted]

**Response to Reviewer#2:**

Brazil is the world's largest sugarcane producer, and accurate estimates of the distribution, harvest area of sugarcane are crucial for sustainable sugarcane production and national food security. The study "High-resolution map of sugarcane cultivation in Brazil using a phenologybased method" aims to identify the harvest area of sugarcane in Brazil by a phenology-based method (TWDTW). The annual 30-m spatial resolution sugarcane harvest maps (2016-2019) for 14 states in Brazil have been generated with very high accuracy (over 98%).

Thank you for your comments. Your suggestions are very useful for us to improve our research. We revised our manuscript accordingly. The changes in our manuscript are underlined with red. We believe our manuscript improved a lot after the modification. Please see the response below.

**In terms of the method, I have some concerns.**

**1. (a) The field sample data.**

**The authors used the Google Earth images to select the sugarcane samples without any ground truth samples as references, so it is difficult to tell the crop types. Meanwhile, the authors did not describe the rules of visual interpretations. It is hard to tell the accuracy of the sugarcane samples selected.**

Thanks for your question. It is a pity that we have no ground truth samples to validate the maps. But we have strict rules in our sample selection process: (1) selected samples using visual interpretations according to color and textures of the images from Google Earth, and (2) cross checked these samples using the NDVI series curves.

In the revised manuscript, we added a figure to showed the color and textures, and the timeseries of NDVI for sugarcane and non-sugarcane. And we rewrote this section to describe the rules in detail.

"The sample data used for calibration and validation in this study were mainly obtained based on the high-resolution images from Google Earth. We selected the samples for sugarcane and non-sugarcane according to the following rules. First, we selected the samples for sugarcane or non-sugarcane using visual interpretations according to color and textures of the images from Google Earth. As shown in Fig. 3a, sugarcane exhibits unique color and textures on the high resolution images from Google Earth. Sugarcane has coarser surface than most of the crops and smoother surface than forest or trees in growing season, which can be used to separate sugarcane from these types. Second, we cross checked and

confirmed each of these samples using its NDVI timeseries. Sugarcane has a long life cycle ranging from 12 to 18 months (Rudorff et al., 2010), which is longer than most of the crops (Fig. 3b). And the sharply decreased NDVI in sugarcane harvest period can separate sugarcane from forest and pasture (Fig. 3b). We only use the samples which can satisfy the above two criteria. Finally, we collected totally 2909 samples with 1393 for sugarcane and 1516 for non-sugarcane in the year 2018 (Fig. 1)." (Lines 120-132 in the revised manuscript)

[Figure]

Figure 3. Examples of the (a) color and textures on the high-resolution images from Google Earth, and (b) timeseries of NDVI for different vegetations.

**(b) In addition, the NDVI curves were used to select sugarcane samples, but later the selected sugarcane samples were used to extract the standard sugarcane NDVI curves to develop algorithms for sugarcane mapping. Therefore, I am confused that there was a little self-proof and not sure about the accuracy estimated based on these samples.**

Thanks for your consideration. We used 50 samples randomly selected from all the 1393 sugarcane samples for training to produce the standard NDVI curve for sugarcane (Lines 156-158 in the original manuscript). In the revised manuscript, we used the remaining 1343 sugarcane samples and 1516 non-sugarcane samples for validation.

Lines 156-158 in the original manuscript "In Fig. 4, the standard NDVI curves for sugarcane were generated by randomly selecting 50 sugarcane samples across Brazil from the field data in 2018 (Section 2.2.2) and calculating their averaged NDVI values in the same growing period."

**(c) As these samples were selected by Google Earth images rather than filed truth data, it is not reasonable to call these samples as "filed data".**

Thank you, according to your comments, we have changed the "filed data" to "sample data".

**2. The TWDTW method for sugarcane mapping**

**In this work, the "difference" between the maximum NDVI value in the growing season (NDVI$_{max}$: mean value of the two maximum NDVI in the growing season) and the minimum NDVI value in the non-growing season (NDVI$_{min}$: mean value of the two minimum NDVI in the non-growing season) was calculated for each pixel. The "difference" as a criterion to map the sugarcane in Brazil. The evidence was shown as "the differences in NDVI between the growing season and non-growing season for these vegetations were lower than those for sugarcane" in lines of 177-178. Here, the vegetations compared mainly include grassland, seasonal forest, and grazing areas as shown in line of 175.**

**My main concern is that the difficulty of sugarcane mapping is to separate sugarcane from other crop types within agricultural system rather than vegetations in other ecosystems such as grassland and forest. This evidence cannot prove the criterion used in the sugarcane mapping is robust.**

In our study, TWDTW is the main method, and the "difference" the reviewer mentioned here is just a complement to the method to further separate some easily confusable types by using TWDTW (such as pasture and seasonal forest). Most of the crops have growing seasons less than a year which can be directly separated from sugarcane using the TWDTW. As described in the previous study, the lowest performance

in classification sugarcane was observed mainly in regions dominated by pasture, which has similar temporal-spectral behaviour to sugarcane (Xavier et al., 2006). As is shown in Fig.3, the "difference" for pasture and seasonal forest were lower than sugarcane (Ferreira et al. 2004; Mueller et al. 2015). Therefore, we used the NDVI "difference" between the growing season and non-growing season as another supplement criterion to separate sugarcane. You can see the detailed explanation in the revised manuscript:

"(2) Remove the influence of other vegetation with similar NDVI changes to sugarcane harvesting period, such as the Brazilian Cerrado biomes and pasture. In the Brazilian Cerrado, some vegetations (such as grassland, shrubland, woodland, and deciduous forest) exhibited low NDVI values from the end of the drought season to the beginning of the rainy season (August to October), which is similar to the sugarcane harvesting practice. However, the "difference" in NDVI between the growing season and non-growing season for these vegetations were lower than those for sugarcane (Ferreira et al. 2004; Mueller et al. 2015). Therefore, we used the NDVI "difference" as another supplement criterion to further separate sugarcane from the aforementioned vegetation types across the states related to the Brazilian Cerrado (i.e., São Paulo, Goiás, Minas Gerais, Mato Grosso do Sul, Mato Grosso, and Bahia). The "difference" between the maximum NDVI value in sugarcane growing season (NDVI$_{max}$: mean value of the two maximum NDVI in the growing season) and the minimum NDVI value in sugarcane non-growing season (NDVI$_{min}$: mean value of the two minimum NDVI in the non-growing season) was calculated for each pixel in these states (Fig. 7). From the statistics of 50 randomly selected sugarcane samples, we found the "difference" for sugarcane is mostly greater than 0.31. Therefore, we set the pixels with "difference" less than 0.31 as non-sugarcane. Additionally, pasture, which has similar temporal-spectral behaviour to sugarcane (Xavier et al., 2006), was further removed using the pasture maps (overall accuracy of 87%) produced by Parente et al. (2017). Across all the 14 studied states, pixels consecutively labelled as pasture on the pasture maps from 2016 to 2019 was set as non-sugarcane." (Lines 199-206 in the revised manuscript)

**3. The "distance" and "difference" thresholds for identifying sugarcane**

(a) See lines 185-190, the agricultural statistical harvest area for sugarcane at the state and municipality levels have been used to determine the "distance" threshold and optimize the "difference" threshold, however, the statistical data were used to compare with the resultant sugarcane areas. To some extent, this approach is self-proof.

"Lines 185-190: Second, we used the agricultural statistical harvest area for sugarcane at the state level to determine the "distance" threshold. A pixel with "distance" value lower than the "distance" threshold was considered a "sugarcane" pixel, and the total area of all sugarcane pixels should be equal to the statistical harvest area of sugarcane in the investigated state. (4) We optimize the T$_{ND}$ value according to the identification error by comparing the estimated area with the agricultural statistical data at the municipality level. In our study, municipalities with small areas of planted

**sugarcane (less than 1000 ha or less than 1% of the total sugarcane area in the investigated state) were identified separately with other municipalities to improve the identification accuracy of the entire investigated state."**

Thank you for pointing out this. In the revised manuscript, we found most of the sugarcane samples had higher "difference" than 0.31 from 50 randomly selected sugarcane samples. Therefore, we set the "difference" to 0.31 and did not use the agricultural statistical area data at the municipality level in method development. We only used the agricultural statistical area data at state level to determine the "distance" threshold, and used the agricultural statistical area data at the municipality level in validation. Besides agricultural statistical data, we also used 2859 samples in 2018 to validate the sugarcane maps.

The modification in method section is shown as following:

"(2) Remove the influence of other vegetation with similar NDVI changes to sugarcane harvesting period, such as the Brazilian Cerrado biomes and pasture. In the Brazilian Cerrado, some vegetations (such as grassland, shrubland, woodland, and deciduous forest) exhibited low NDVI values from the end of the drought season to the beginning of the rainy season (August to October), which is similar to the sugarcane harvesting practice. However, the "difference" in NDVI between the growing season and non-growing season for these vegetations were lower than those for sugarcane (Ferreira et al. 2004; Mueller et al. 2015). Therefore, we used the NDVI "difference" as another supplement criterion to further separate sugarcane from the aforementioned vegetation types across the states related to the Brazilian Cerrado (i.e., São Paulo, Goiás, Minas Gerais, Mato Grosso do Sul, Mato Grosso, and Bahia). The "difference" between the maximum NDVI value in sugarcane growing season ($NDVI_{max}$: mean value of the two maximum NDVI in the growing season) and the minimum NDVI value in sugarcane non-growing season ($NDVI_{min}$: mean value of the two minimum NDVI in the non-growing season) was calculated for each pixel in these states (Fig. 7). From the statistics of 50 randomly selected sugarcane samples, we found the "difference" for sugarcane is mostly greater than 0.31. Therefore, we set the pixels with "difference" less than 0.31 as non-sugarcane. Additionally, pasture, which has similar temporal-spectral behaviour to sugarcane (Xavier et al., 2006), was further removed using the pasture maps (overall accuracy of 87%) produced by Parente et al. (2017). Across all the 14 studied states, pixels consecutively labelled as pasture on the pasture maps from 2016 to 2019 was set as non-sugarcane. (3) Identify to produce the sugarcane maps. We used the agricultural statistical harvest area for sugarcane at the state level to determine the "distance" threshold. A pixel with "distance" value lower than the "distance" threshold was considered a "sugarcane" pixel, and the total area of all sugarcane pixels should be equal to the statistical harvest area of sugarcane in the investigated state." (Lines 199-225 in the revised manuscript)

**(b) What's more, statistical data cannot present the spatial patterns of the sugarcane fields. Therefore, I doubt the ability of this approach to describe the spatial information of the sugarcane**

**distribution in Brazil.**

The validation using statistical data can represent the accuracy at the spatial in some ways. On the one hand, the administrative regions at the municipality level are so small (with an averaged area of 500 ha) that they can represent the spatial distribution information (Fig.4; Fig.10). And the detailed statistical data have more superiority to express spatial distributions than sample data. The detailed statistical data contains sugarcane information all over the study area, while the sample data contains the information at their locations. On the other hand, our method was developed by allocating the pixels to sugarcane and non-sugarcane according to the similarity and the harvest area of sugarcane. Pixels with low distance indicate high similarity and a high probability of being associated with the sugarcane. The method works based on the spatial similarities therefore contains spatial information. What's more, the validation showed the sugarcane maps were in high accuracy at both pixel level and regional level. Therefore, the maps produced in our study can exhibit the spatial distribution of sugarcane well.

[Figure]

Figure 4. The administrative boundaries for (a) municipalities, (b) microregions, and (c) mesoregions. The boundary lines at the municipality and state levels were downloaded directly from the IBGE, and we aggregated the municipalities into microregions and mesoregions according to the regions in PAM denoted by the IBGE.

[Figure]

Figure 10. Comparisons between the agricultural statistical harvest area and the estimated harvest area of sugarcane at the municipality level in Brazil in 2018. "N" and "mean" represents the number and mean value of all the valid statistical data in each figure, respectively.

**Minor:**

**1. Time-weighted dynamic time warping (TWDTW) method**

**TWDTW is the method used in this work to map sugarcane, but the description of the method is not detail.**

Thank you. We added more detailed description of TWDTW in the revised manuscript.

"TWDTW is a time-weighted version of the DTW method for land use and land cover classification (Belgiu et al., 2018; Dong et al., 2020a). The DTW works by comparing the similarity between two sequences, such as the unknown sequence (Y) and target sequence (X), through warping the unknown sequence (Y) to search for their optimal path and obtain their minimum distance, namely the similarity (dissimilarity) (Petitjean et al., 2012; Petitjean and Weber, 2014). The TWDTW method adds a temporal cost accounting for the phase difference between the two time series to the minimum distance (Maus et al., 2016). The TWDTW algorithm includes three steps in land use classification and identification: (1) generate the standard curve of a selected index (e.g., NDVI) for several crops or a single crop (e.g., sugarcane) based on time series images and field sample training data; (2) find the best alignment, and generate the dissimilarities (TWDTW distances) between unknown curves of NDVI time series (i.e., the NDVI curves of the unknown pixels) with the standard NDVI curve of sugarcane; (3) identify the unknown pixels based on the TWDTW distance; in this process pixels with low distance indicate high similarity and a high probability of being associated with the specified class (i.e., sugarcane)." (Lines 150-163 in the revised manuscript)

**2. Sugarcane in Brazil covers an extensive harvesting period, how were all the possible standard NDVI curves collected in Figure 5?**

[Figure]

**Figure 5 (Figure 7 in the revised manuscript). Seasonal changes in the NDVI series for sugarcane in the south-central and northeast Brazil. The gray areas are the periods used to calculate the maximum NDVI value in growing season ($NDVI_{max}$) and the minimum NDVI value in non-growing season ($NDVI_{min}$).**

Sugarcane in Brazil covers an extensive harvesting period (Rudorff et al. 2010). Generally, sugarcane can be harvested from April to December in the south-central area, and from September to April in the next year in the northeast areas. Therefore, Fig. 5 (Figure 7 in the revised manuscript) was generated by repeating the standard NDVI curve in Fig. 4 (Figure 6 in the revised manuscript) to ensure all the standard NDVI curves cover the growing period of sugarcane at 16-day interval for both south-central areas (April to December) and northeast areas (September to April in the next year). The TWDTW method works by comparing the similarities between of two time series that can tolerate phenological changes, so there is no need and impossible to fill all the gaps between each standard curve in Fig. 5. Therefore, the standard NDVI curves are sufficient to represent the sugarcane behavior in most places in Brazil.

[Figure]

Figure 4 (Figure 6 in the revised manuscript). Growing stages of sugarcane and the respective NDVI changes. The grey and red symbols represent the NDVI curve for the planted sugarcane with a 12–18 month cycle, and the green and red symbols represent the NDVI curve for the ratoon sugarcane with a 12 month cycle. The growing stages of the 12 month cycle sugarcane (germination, tillering, grand growth, and ripening) are labeled in the figure.

**3. It is needed to provide more details about the method. In addition, more descriptions are needed about the flexible phenological and harvest conditions of sugarcane in the TWDTW in lines of 165-167.**

**"Lines 165-167: According to the phenology of sugarcane in the south-central and northeast areas, Figure 5 shows the possible standard NDVI curves (these NDVI curves were repeated from the NDVI standard curve for sugarcane, as denoted by the red symbols and line in Fig. 4) for sugarcane in south-central and northeast Brazil. In this study, we incorporate the flexible phenological and harvest conditions of sugarcane in Brazil into the TWDTW method as follows."**

Thank you. We added more details about the method.

(1) We added more detailed description of TWDTW method: see the response to the comment#1 in minor.

(2) We added more detailed description about sample selection (response to the comment#1 in major concern).

(3) We added a flowchart in the method section (2.3.2 Employing the TWDTW method for sugarcane mapping) to describe the application more clearly.

"The workflow by employing the TWDTW method for sugarcane harvest area mapping are shown in Fig.5." (Line 173 in the revised manuscript)

[Figure]

Figure 5. The workflow by employing the TWDTW method for sugarcane harvest area mapping.

"the flexible phenological and harvest conditions of sugarcane" means the difference of planted time, growing season length, and harvest time of sugarcane in different fields and regions. We have described it using three paragraphs before Lines 165-167 in the original manuscript.

Growing stages of sugarcane:

"The growing period of sugarcane can be separated into four phases: the germination, tillering, grand growth, and ripening, and the NDVI values vary in different growing phases (Fig. 4). (1) Germination phase. Sugarcane begins to germinate approximately 15-30 days after planting. The NDVI starts to increase in this period. (2) Tillering phase. Tillering phase starts after approximately 2 months of

germination, and tillers emerge from the base of the mother shoot to form 5-10 stalks. The NDVI increases quickly in this period. (3) Grand growth phase. This phase spans a period of approximately 4-10 months after planting. The NDVI peaks during this period. (4) Ripening and harvesting phase. In this phase, the NDVI starts to decrease, and the moisture content in sugarcane drastically drops." (Lines 145-151 in the original manuscript)

Growing season length of sugarcane:

"From planting until the first cut, the crop is called planted sugarcane, and the growth cycle lasts between 12 and 18 months depending on the season, variety, and region of planting. After the first harvest, ratoon sugarcane is harvested yearly with a normal cycle of 12 months for a period of approximately 5 to 7 years or more (Rudorff et al., 2010). Although planted sugarcane and ratoon sugarcane have different length in growing cycles, they shared similar NDVI curves from the grand growth to ripening phases (red symbols in Fig. 4) which can be used as the standard seasonal curve for harvest area mapping. In Fig. 4, the standard NDVI curves for sugarcane were generated by random selecting 50 sugarcane samples across Brazil from the field data in 2018 (Section 2.2.2) and calculating their averaged NDVI values in the same growing period." (Lines 152-158 in the original manuscript)

Harvest time of sugarcane in different regions:

"Sugarcane in Brazil covers an extensive harvesting period (Rudorff et al. 2010). In the south-central area (including São Paulo, Goiás, Minas Gerais, Mato Grosso do Sul, Paraná, Mato Grosso, Bahia, Rio de Janeiro, and Espírito Santo), sugarcane is often harvested from April to December, with a harvesting season spanning 9 months. In the northeast area (including Alagoas, Pernambuco, Paraíba, Rio Grande do Norte, and Sergipe), sugarcane is harvested from September to April in the next year, with a harvesting season spanning 8 months. According to the phenology of sugarcane in the south-central and northeast areas, Figure 5 shows the possible standard NDVI curves (these NDVI curves were repeated from the NDVI standard curve for sugarcane, as denoted by the red symbols and line in Fig. 4) for sugarcane in south-central and northeast Brazil. In this study, we incorporate the flexible phenological and harvest conditions of sugarcane in Brazil into the TWDTW method as follows." (Lines 160-167 in the original manuscript)

**4. In line 170, how the possible "distance" values have been calculated? Please give more explanations here.**

**Line 170 "(1) We calculated all the possible "distance" values for each pixel by comparing the seasonal changes of NDVI curve with the standard curves of sugarcane (Fig. 5) based on the TWDTW method from 2016-2019. In this process, we can obtain multiple "distance" values for each pixel, and we selected the minimum value of the "distance" as the distance value for each pixel."**

Thank you. The possible "distance" values have been calculated by comparing the NDVI timeseries for each pixel with the standard NDVI curves of sugarcane using the TWDTW method. For there are 13 standard NDVI curves for sugarcane, we obtain 13 distances for each pixel. We rewrite these sentences to express more clearly.

"(1) For each year from 2016-2019, we calculated all the "distance" values for each pixel by comparing its NDVI timeseries with the standard NDVI curves of sugarcane in south-central area and northeast area (Fig. 7) based on the TWDTW method. In this process, we can obtain 13 "distance" values corresponding to the 13 standard NDVI curves in Fig. 7 for each pixel, and we selected the minimum value of the "distance" as the final distance value for each pixel." (Lines 202-205 in the revised manuscript)

---

## Author Response (AR2)

**Journal:** ESSD

**Title:** High-resolution map of sugarcane cultivation in Brazil using a phenology-based method

**MS No.:** essd-2021-88

**MS Type:** Data description paper

Dear Prof. Kirsten Elger,

We are very grateful to you for your time and effort to our manuscript "High-resolution map of sugarcane cultivation in Brazil using a phenology-based method" (MS No.: essd-2021-88). Our manuscript can not be processed so smoothly without your help.

We revised our manuscript according to your comments and the remarks from the preceding review file validation by the editorial support team. Please find the point-by-point responses below. Please note that the comments from you and the editorial support team are in **bold** followed by our responses in regular text. The changes in our manuscript are underlined with red.

We believe the quality of the manuscript can now meet the high standard of ESSD and deeply appreciate your consideration of our manuscript.

Sincerely,

Yi Zheng, Wenping Yuan

School of Atmospheric Sciences,

Sun Yat-sen University, Zhuhai 519082, Guangdong, China

Email: yuanwpcn@126.com

**Response to Prof. Kirsten Elger:**

**Dear Yi Zheng and co-authors,**

**many thanks for your excellent revision of the manuscript. Before finally accepting it for publication in ESSD, I wanted to ask you for two minor corrections to the manuscript:**

**New Figure 8 (p. 24 in the track change mode version): As one can only see sugarcane (in red), I would remove the "non-sugarcane" field of the legend here and**

**add this legend to Figure 9. Here the high resolution images allow for distinguishing presence and absence of sugarcane. Alternatively, you could also describe the "presence (red) and absence (white)" in the figure caption.**

**many thanks and best regards,**

**Kirsten Elger**

Dear Prof. Kirsten Elger, thank you very much for your positive comments to our revision of the manuscript. As your comments, we removed non-sugarcane in Figure 8 and added description about "presence (red) and absence (white)" to the caption of Figure 9. Please see Figure 8 and Figure 9 below:

[Figure]

Figure 8. Sugarcane harvest map for the 14 studied states in Brazil in 2018. The administrative boundary data were obtained from the IBGE.

[Figure]

Figure 9. Zoomed-in of (a1)-(d1) high-resolution images from Google Earth © Google Earth, (a2)-(d2) presence (red) and absence (white) of sugarcane on the harvest maps in 2018 for the typical area A-D in Fig. 8.

**Response to the remarks from the preceding review file validation:**

**1. Regarding your figures #3, #9: with the next revision, please add the copyright icon as follows: ©
Google Earth.**

Thank you. As your comments, we added copyright icon "© Google Earth" to the caption of Figure
3 and Figure9.

"Figure 3. Examples of the (a) color and textures on the high-resolution images from Google Earth
© Google Earth, and (b) timeseries of NDVI for different vegetations."

"Figure 9. Zoomed-in of (a1)-(d1) high-resolution images from Google Earth © Google Earth, (a2)-
(d2) presence (red) and absence (white) of sugarcane on the harvest maps in 2018 for the typical area A-
D in Fig. 8."

**2. Please ensure that the colour schemes used in your maps and charts allow readers with colour
vision deficiencies to correctly interpret your findings. Please check your figures using the Coblis –
Color Blindness Simulator (https://www.color-blindness.com/coblis-color-blindness-simulator/)
and revise the colour schemes accordingly.**

Thank you for pointing out this. We agree that it is very important to improve the accessibility of
figures for readers with colour vision deficiencies. We checked all the figures in our manuscript using the
Coblis – Color Blindness Simulator and revised Figure 2–3, Figure 6–7, and Figure 12–14 accordingly.
We think all the figures in our manuscript can be easily and correctly interpreted by all the readers now.
Please tell us if there are any remaining mistakes in these figures.